

# A Prototype Passive Microwave Retrieval Algorithm for Tundra Snow Density

Jeffrey J. Welch & Richard E. J. Kelly

Geography and Environmental Management, University of Waterloo, Waterloo, Canada

*Correspondence to*: Jeffrey J. Welch (jjwelch@uwaterloo.ca)

**Abstract.**

Snow density data are important for a variety of applications, yet, to our knowledge, there are no robust methods for estimating spatiotemporal varying snow density in the Arctic environment. The current understanding of snow density variability is largely limited to manual in situ sampling, which is not feasible across large domains like the Canadian Arctic. This research proposes a passive microwave retrieval algorithm for tundra snow density. A two-layer electromagnetic snowpack model, representing depth hoar underlaying a wind slab layer, was used to estimate microwave emissions for use in an inverse model to estimate snow density. The proposed algorithm is predicated on solving the inverse model at boundary conditions for the snowpack layer densities to estimate snow density within a plausible range. An experiment was conducted to assess the algorithm's ability to reproduce snow density estimates from snow courses at four high arctic sites in the Canadian tundra. The electromagnetic snowpack model was calibrated at one site and then evaluated at the three other sites. Results from the calibration and evaluation sites were similar and the algorithm replicated the density estimates from snow courses well with absolute error values approaching the uncertainty of the reference data (±10%). The algorithm configuration appears best suited for estimating snow density conditions towards the end of the winter season. With more extensive forcing data (e.g. from global climate models) this algorithm could be applied across the tundra to provide information on snow density at scales that are not currently available.

## 1 Introduction

There are numerous applications for which the quantification of snow density is important: for example, estimating snow water equivalent (SWE) for water resources (Venäläinen et al., 2021, 2023), modelling atmosphere-land interactions for energy balances (Gouttevin et al., 2012, 2018), and ecological monitoring of Arctic fauna (Martineau et al., 2022; Sivy et al., 2018); though, to the best of our knowledge, there is no robust method for estimating spatiotemporally-varying snow density in the Arctic. There are automated instruments to estimate snow density but they are not widely implemented, instead density is typical estimated by weighing a known volume of snow (Kinar & Pomeroy, 2015). This manual process is labour intensive and, as a result, sparsely distributed making the prediction of spatially distributed density estimates uncertain. In a remote environment, like the Canadian Arctic, comprehensive in situ sampling is not feasible due to logistical constraints, so large-scale analyses involving snow density tend to rely on modelled estimates. Recent studies have shown that current snow density





products, from meteorological reanalysis or detailed snow models, are not adequate for use in Arctic environments. The snow scheme in the ERA5-Land reanalysis model overestimates snow depth and underestimates density, by considerable margins, in high-latitudes (Cao et al., 2020, 2022). Similarly, detailed snow models (i.e. Crocus and SNOWPACK) cannot estimate the expected vertical density profile in the Arctic (Barrere et al., 2017; Domine et al., 2019). Despite its intrinsic importance in

Earth systems, snow density variability is currently not well understood on large spatiotemporal scales.

One possible approach to estimate snow density at the regional scale (i.e. $10^2$-$10^4$ km²; Woo, 1998) is from satellite-based remote sensing. Satellite passive microwave (PM) radiometry offers near-daily coverage of the Northern Hemisphere, under most weather conditions, with a data record spanning back to 1978. Emitted microwave energy can pass through a snowpack unattenuated at lower frequencies or is attenuated at higher frequencies. For attenuated emission, the primary microwave

interaction within a dry snowpack is volume scattering which is controlled by the snowpack properties (i.e. snow depth, density, temperature, and grain size radius; Chang et al., 1982). PM snow emission retrievals using a frequency difference approach (ΔTb) – the subtraction of higher frequency channel Tb (volume scattering dominated) from a lower-frequency Tb channel (subnivean emission dominated) – have been the basis of empirical representations of PM estimates (e.g. Chang et al., 1987) and more sophisticated assimilation-based retrieval schemes (e.g. Takala et al., 2011). Historically, snow mass has been

estimated with spaceborne (PM) radiometry through retrieval algorithms focusing on snow depth (Kelly et al., 2003, 2019; Takala et al., 2011; Tedesco & Jeyaratnam, 2016). In theory, the principles behind those existing retrieval schemes could be exploited to estimate snow density rather than depth.

In general, the parameterization of snow density in has been simplified in large-scale passive microwave SWE estimation models (Mortimer et al., 2022). There is a lack of snow density observations at the necessary scales to constrain density

parameterization, primarily because of the difficult in acquiring spatially distributed in situ observations (Sturm et al., 2010). As a result, snow depth has been the focus of most analyses regarding SWE. In some cases, snow density is kept constant across the domain (e.g. Luojus et al., 2021; Takala et al., 2011) or conservative estimates are taken from empirical models of snow density evolution over time (e.g. Kelly et al., 2003). However, such a simplified representation of snow density may not adequately represent variability across the large domains those models are designed to cover.

In this study, an experiment was conducted to evaluate the potential use of satellite-based PM observations and existing in situ meteorological networks to estimate snow density in the high Arctic tundra biome. Snow density estimates from the proposed algorithm could provide a notable benefit over existing snow density products, which do not account for the proper snow densification schemes relevant to the tundra environment (Cao et al., 2022; Domine et al., 2016). Instead, the algorithm would be informed by independent PM observations that provide context on snow density conditions and not rely on the

parameterization of specific densification schemes. Thus, estimates from this approach could fill a gap in the understanding of snow density variability in remote areas that are unsuitable for in intensive in situ sampling and where current snow density models are not appropriate.




## 2 Study Area

Four automatic weather stations (AWS) were identified across a latitudinal range in the Canadian tundra for this experiment
(Fig. 1); statistical summaries for each site are provided in Table 1. These sites were specifically selected because they are
located in the high Arctic tundra environment and collocated with manual in situ sampling sites (described in Section 3.2). The
tundra environment was chosen to develop this prototype snow density retrieval algorithm for the following two key reasons
that tend to simplify the retrieval process. First, terrain effects should be minimal compared to those found in more
topologically complex landscapes like alpine environments (Tong et al., 2010). Second, forest cover attenuation effects (Li et
al., 2020) are minimized in tundra regions which are characterized by sparse, short vegetation (Marsh & Pomeroy, 1996).

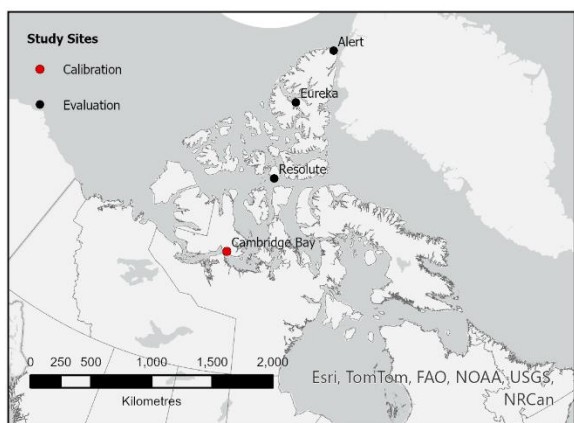

**Figure 1 – Study sites (AWS), distributed across the high Arctic tundra in Nunavut, Canada.**

**Table 1 – Statistical summaries of study sites: average AWS data (Jan-Mar) and CanSWE reference snow density data.**

| Site | Latitude | AWS Data | | CanSWE Density Data (kg/m³) | | | |
|---|---|---|---|---|---|---|---|
| | | Avg. Temp (C) | Avg. SD (cm) | n | Avg. | Std. | Min/Max |
| Alert | 82°31' | -30.2 | 31.7 | 64 | 356.5 | 49.9 | 147/440 |
| Eureka | 79°59' | -35.2 | 17.4 | 80 | 329.4 | 55.2 | 143/436 |
| Resolute | 74°43' | -29.4 | 19.4 | 56 | 366.1 | 55.2 | 243/485 |
| Cambridge Bay | 69°06' | -30.6 | 28.4 | 229 | 323.9 | 45.2 | 185/452 |

## 3 Data

### 3.1 Model Forcing Data

PM radiometry data were the main forcing for the proposed snow density retrieval algorithm. Radiometry data were
acquired from the Advanced Microwave Scanning Radiometer Earth Observing System (AMSR-E) Level-2A product gridded
to a 25x25 km Equal-Area Scalable Earth (EASE) grid (Ashcroft & Wentz, 2013), spanning eight winter seasons (2003-2011)
while the instrument was functional (reference snow density data were not available for the 2002-03 season). AMSR-E
observations for each station were extracted from an adjacent EASE grid cell to the AWS to minimize water fraction in
observation scene due to their proximity to the coast. Nighttime observations from the descending orbit track (~1:30 am local
time at the equator) were used so snow conditions would be more likely to be cold and dry for optimal microwave retrievals
(Derksen et al., 2005). The 18.7 and 36.5 GHz vertically-polarized radiometer channels (hereafter 19 and 37 GHz, respectively)
were used to estimate ΔTb in the forward model.



Meteorological measurements, acquired from the Environment and Climate Change Canada (ECCC) AWS network (ECCC
& ClimateData.ca, n.d.) were also used for model forcing. The electromagnetic snowpack model was parameterized with AWS
data, which required daily measurements of snow depth and air temperature as prior snow conditions. AWS data were the
limiting factor in this experiment because the AWS network is sparsely distributed in northern Canada limiting potential study
sites.

## 3.2 In situ Reference Data

The curated ECCC Canadian Historical Snow Water Equivalent dataset (CanSWE; Vionnet et al., 2021) provided in situ
snow density data for comparison with algorithm estimates. CanSWE included sampling locations collocated with AWS sites
which allowed for direct comparisons of estimated and sampled snow density. Snow density data in CanSWE (considered in
this study) were collected with ESC-30 SWE tubes along 5-10 point snow course transects spanning 150-300m, aggregated
into bulk estimates of snow density. A ten percent uncertainty range was applied to the snow density data in the reference
dataset because of uncertainties inherent to manual snow density sampling (Conger & McClung, 2009; López-Moreno et al.,
2020). Specific information about sampling procedures was not available for the individual sites in the CanSWE dataset (e.g.
where the snow course is situated relative to the AWS was unknown).

CanSWE snow density data from four manual sampling sites were used in the development of this algorithm. Those data
were chosen specifically because of their location in the high Arctic tundra with relatively high temporal sampling frequency.
The reference dataset was limited with respect to the algorithm configuration (described in Section 4.2). A number of yearly
AWS forcing datasets were deemed unsuitable for algorithm forcing and were removed from the analysis. One winter season
at the Eureka site (2008-09) had insufficient snow accumulation to permit PM retrievals (i.e. <10 cm) and three seasons each
for Alert (2007-08, 2009-10, and 2010-11) and Resolute (2003-04, 2004-05, and 2006-07) where snow accumulation
trajectories reported by the AWS were starkly different from the in situ snow depth samples in CanSWE. Individual CanSWE
snow density samples were removed under three conditions: if they were out of the domain of algorithm estimates (i.e. 150-
450 kg/m$^3$, described in Section 4.2), sporadic observations that did not fit temporally with the seasonal trajectory, and high
densities late in the season during ablation when the snowpack would likely be in a wet state inhibiting microwave emissions.

## 4 Methods

### 4.1 Electromagnetic Model

The Snow Microwave Radiative Transfer model (Picard et al., 2018), configured with the Dense Media Radiative Transfer
(DMRT) electromagnetic model, was used in this study. The physically-based forward modelling approach required the
snowpack to be parameterized, so the relevant characteristics needed to be quantified. A two-layer snowpack model was
configured to account for the presence of depth hoar underneath a slab layer to best represent the microwave signature of



tundra snow (Hall, 1987; Saberi et al., 2017). Upon initial deposition the snowpack would likely be in a homogenous state, with one layer, but that situation was not considered in this approach. The strong environmental controls present in the tundra

contribute to the development of wind slab and depth hoar snow layers quickly after deposition (Benson & Sturm, 1993; Sturm & Holmgren, 1998), and algorithm retrievals were performed after 10 cm of snow had accumulated so the pack would be unlikely to be in the initial homogenous state. The snow depth forcing variable was prescribed by dividing the total snow depth at the AWS into the relative depths for the two layers using a fixed 1:2 ratio of depth hoar to slab layer thickness (Saberi et al., 2017), representative of high Arctic tundra snow on a regional-scale (Derksen et al., 2014; Meloche et al., 2022). Similarly,

the minimum daily air temperature at the AWS was used as a surrogate for snow temperature and was prescribed directly to each snow layer.

The microstructure model in SMRT (sticky-hard-spheres) required estimates of the effective radius of ice grains in the snowpack which are not acquired by operational AWS measurements. The effective grain radius model from Kelly et al. (2003) was modified for use with the two layer snowpack model – Sturm & Benson's (1997) kinetic "lower" grain growth model was

applied to the depth hoar layer and a slow, constant growth rate was applied to the upper wind slab layer to represent equilibrium growth. The grain growth model required parameterization of the minimum and maximum grain radius which were determined through a calibration routine (see Section 3.5).

The electromagnetic model used in this study included simplified substrate and atmospheric components. Given the cold temperatures of the study area, the substrate was assumed to consist of frozen soil, so the effects of dielectric permittivity and

roughness should be negligible when estimating $\Delta Tb$ (Kelly et al., 2003). The substrate composition was parameterized to represent cryosolic soil, the predominant soil type found in the Canadian Arctic (Tarnocai & Bockheim, 2011). AWS observations of minimum daily temperature +5C were used to parameterize the substrate temperature because of the insulative properties of snow (Benson & Sturm, 1993). Atmospheric contributions were not considered. A full list of model parameters is provided in Table 2.

**Table 2 – Electromagnetic model parameterization.**

| Snowpack Model | | Substrate Model | |
|---|---|---|---|
| Parameter | Value | Parameter | Value |
| Electromagnetic Model | Dense Media Radiative Transfer based on Quasicrystallin Approximation with coherent potential (Tsang et al., 2000) | Composition | Cryosolic soil as described by Tarnocai & Bockheim (2011): Sand - 75%, Clay – 8%, Dry matter – 1490 kg/m³ |
| Snow Depth | AWS snow depth portioned in 1:2 ratio of depth hoar to wind slab (Saberi et al., 2017) | Temperature | AWS minimum daily 2m air temperature + 5C |
| Temperature | AWS minimum daily 2m air temperature | Permittivity Model | Dobson et al. (1985) |
| Grain Radius | Modified growth model from Kelly et al. (2003) | Roughness | Flat surface (i.e. no surface roughness) |
| Stickiness | Non-sticky spheres (i.e. infinite stickiness) | Moisture Content | <1% |
| Liquid Water Content | 0% | | |
| Salinity | 0% | | |



## 4.2 Snow Density Retrieval Algorithm

The GlobSnow grain size estimation procedure – using snow depth measurements from AWS to optimize the effective snow grain size parameter in the emission model (Pulliainen, 2006; Takala et al., 2011) - was modified to produce estimates of snow density. PM retrievals of snow density were conducted at each AWS site, where meteorological conditions dictated

when retrievals were performed. A minimum snow depth of 10 cm was imposed for algorithm retrievals because of the transparent nature of shallow snow to microwave emissions (Hall et al., 2002). Similarly, algorithm retrievals were not conducted when AWS air temperatures were above freezing because of the likelihood of liquid meltwater in the snowpack attenuating microwave emissions (Foster et al., 1984). With the AWS observations prescribed to the electromagnetic model an inverse modelling approach was applied to optimize the snow density parameters. The forward model was inverted by

minimizing the cost function (J):

$$J(\rho_{slab}, \rho_{hoar}) = (\Delta Tb_{sim}(\rho_{slab}, \rho_{hoar}) - \Delta Tb_{obs})^2 \tag{1}$$

representing the vertically polarized 19 and 37 GHz spectral difference in the AMSR-E observation ($\Delta Tb_{obs}$) and the simulated SMRT signature at the same channels ($\Delta Tb_{sim}$), given the prescribed wind slab and depth hoar layer densities ($\rho_{slab}$ and $\rho_{hoar}$, respectively). Algorithm estimates were smoothed with a 5-day moving average to address noise in the radiometry data.

The solution to the two-layer snowpack model presented was imprecise because different layer density combinations could produce the same predicted ΔTb, resulting in a system with no global minima. The practical impact of this equifinality issue was that the algorithm may be confronted by seemingly equally valid but different layer density combinations, producing the same microwave signature. Without additional information there was no suitable way to identify the optimal layer density combination, so the retrieval algorithm was designed to solve for all DMRT-plausible layer density combinations for a given

observation scene to address equifinality in the inverse model.

To constrain the modelled layer density estimates to a plausible range, boundary conditions were established to limit the parameter space in which the algorithm could search for solutions to the inverse model. A lower boundary was defined based on the strong environmental controls present in the tundra that result in a characteristic wind slab snow layer overlaying less dense depth hoar (Benson & Sturm, 1993). The wind slab layer should be denser than the depth hoar layer, so all parameter

combinations where $\rho_{slab} < \rho_{hoar}$ were discarded, and the lower boundary was situated where the two layers had equal snow density values. The upper boundary for the model was defined based on the behaviour of microwave interactions in DMRT. In DMRT theory (in a non-sticky configuration, as applied here) the scattering coefficient for 37 GHz peaks at snow density of 150 kg/m³ and decreases until a volume fraction of 50% (Picard et al., 2013). Thus, the domain of each layer was limited to densities between 150-450 kg/m³ to ensure consistent behaviour in the electromagnetic snowpack model, and the upper

boundary was situated where either layer was at the edge of that domain. An important aspect of the retrieval algorithm was to exploit how the various minima on the cost surface (defined by $J$) were positioned throughout the parameter space. Figure 2 shows an example of how the positions of minima formed a valley transecting the parameter space. Therefore, the DMRT-

plausible layer density combinations were the set of layer density combinations situated along a straight line connecting the solutions at the two established boundary conditions for the inverse model. It should be noted that under some instances, the "valley" intersected with the upper boundary related to the minimum depth hoar density (i.e. left axis in Fig. 2), though the situation shown in Fig 2. (intersecting the upper axis) was more common.

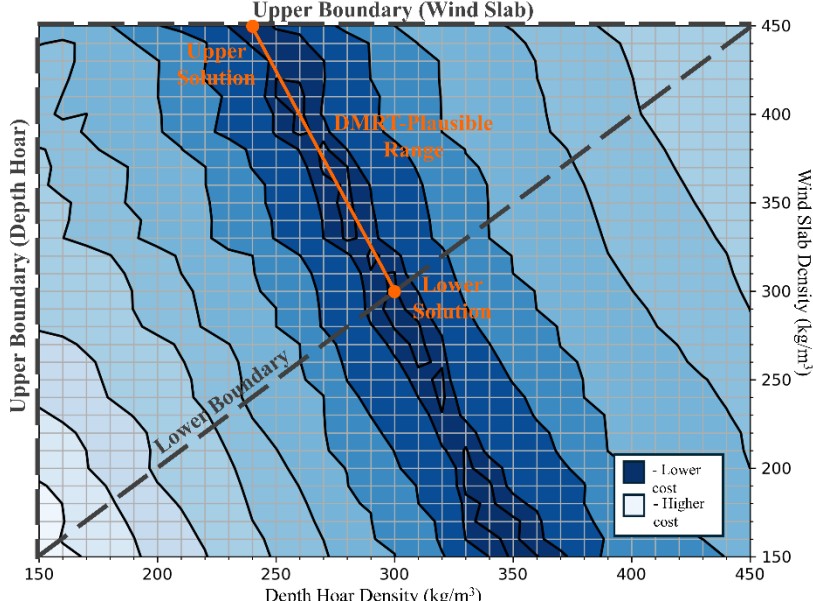

**Figure 2 – Example parameter space (i.e. depth hoar and wind slab layer density combinations), showing upper and lower boundary conditions for snowpack model densities, inverse solutions at the boundaries, and DMRT-plausible range between them. The surface is defined by cost function (J in Eq. 1) where darker (lighter) shades of blue represent lower (higher) cost.**

The range of DMRT-plausible snow densities raised the question of how to evaluate the algorithm estimates against the reference data. A heterogeneity (H) parameter was introduced into the algorithm to estimate densities for the two snow layers and reduce the DMRT-plausible snow densities to a single estimate of bulk snow density – H=0.00 at the lower boundary solution and H=1.00 the upper boundary solution (i.e. the least and most heterogenous solutions, respectively). There did not appear to be any relationship between forcing variables and where the in situ snow density samples were situated within the DMRT-plausible range (and stratigraphic data were not available in the reference dataset), so H was assigned a fixed value determined through a calibration routine. Ultimately, the bulk snow density estimated with H was treated as the final algorithm estimate with uncertainty defined by the DMRT-plausible range.

**4.3 Calibration and Evaluation Procedure**

Some algorithm parameters could not be based on observations and instead needed to be determined through a calibration procedure. The calibration procedure consisted of two stages to identify the optimal algorithm configuration to be applied to all sites over the study period (i.e. 2003-11). First, the values for the minimum and maximum radii in the grain growth model



(described in Section 4.1) were adjusted to produce the greatest overlap between the range of DMRT-plausible snow density estimates and the in situ reference samples, with an overlap metric:

$$overlap = \frac{1}{n} \cdot \sum_{t=1}^{n} \frac{|\{\rho_{est}(t)\} \cap \{\rho_{obs}(t)\}|}{|\{\rho_{est}(t)\}|} \tag{2}$$

where $\{\rho_{est}(t)\}$ is the set of DMRT-plausible estimated snow densities and $\{\rho_{obs}(t)\}$ the set of the corresponding CanSWE density sample with a ±10% uncertainty range, at time $t$. Thus, the overlap metric describes the proportion of the DMRT-plausible snow density range that intersected the uncertainty range of the in situ samples, averaged over $n$ time steps. Second, the value for H (described in Section 4.2) was determined by converting the DMRT-plausible algorithm estimates, from the first step, to minimize the mean absolute percentage error (MAPE) between snow densities and the reference data. MAPE was

chosen for this purpose, rather than absolute or squared error, because of the heteroscedastic nature of the uncertainty in the reference dataset.

The Cambridge Bay AWS site was chosen for the calibration procedure because there were many more CanSWE data available compared to the other AWS sites (Table 1), as it had a shorter sampling interval and forcing data for all winter seasons in the study period. The other three AWS sites were then used to evaluate the calibrated algorithm configuration. At

each site, algorithm snow density estimates were evaluated against the reference snow density samples using the same metrics as in the calibration stage (i.e. overlap and MAPE); bias, root mean square error (RMSE), and correlation (r) were also reported as indicators of algorithm performance. MAPE was treated as the primary measure of absolute accuracy of algorithm estimates; if MAPE was within the uncertainty range of the in situ samples (±10%) then snow density estimates from the algorithm could be comparable to those collected with snow courses.

**Table 3 – Algorithm performance metrics relative to CanSWE reference samples (mean normalized values shown in parentheses).**

| Stage | Site | n | Overlap (%) | MAPE (%) | Bias (kg/m³;%) | RMSE (kg/m³;%) | Correlation |
|---|---|---|---|---|---|---|---|
| **Calibration** | Cambridge Bay | 229 | 39.6 | 13.3 | 9.0 (2.8) | 49.6 (15.3) | 0.426 |
| | Alert | 64 | 42.6 | 14.0 | 32.4 (9.1) | 56.1 (15.7) | 0.547 |
| **Evaluation** | Eureka | 80 | 34.7 | 14.3 | -16.5 (-5.0) | 63.8 (19.4) | 0.382 |
| | Resolute | 56 | 36.5 | 13.5 | 25.6 (7.0) | 54.4 (14.9) | 0.510 |

## 5 Results

### 5.1 Calibration Results – Cambridge Bay

The calibration procedure was applied at the Cambridge Bay site where the algorithm was run for each winter season and the results aggregated to identify the optimal parameter configuration. Performance metrics for the calibrated algorithm are

reported in Table 3. During the first stage of the calibration procedure, the optimal values for the minimum and maximum radii in the grain growth model were 0.30 and 0.90 mm, respectively, and the DMRT-plausible range of estimates overlapped 39.6% with the reference data. In the second calibration stage the optimal value for H was 0.465 and the final estimates of bulk snow density had a MAPE of 13.3%. Over the study period, the algorithm overestimated snow density at Cambridge Bay by a




relatively small amount (9.0 kg/m$^3$; 2.8%) and demonstrated a moderate positive correlation (0.426) with the reference data.

Although, performance over the study period was not consistent and the algorithm configuration performed better during some years than others (Fig. 3). In some cases there was considerable overlap between the algorithm estimated DMRT-plausible snow densities and the reference data and MAPE within reference uncertainty (Fig. 3a). In other cases, algorithm estimates were less skilful earlier in the season then estimates converged closer to the reference samples later on (Fig 3b&c). Overall, calibration results appeared to replicate density estimates from snow courses well with MAPE of final snow density estimates

(converted with H) approaching the level of reference uncertainty and similar magnitudes of algorithm (DMRT-plausible range) and reference uncertainty.

## 5.2 Evaluation Results – All Other Sites

Performance metrics for the evaluation sites were comparable to those achieved at Cambridge Bay during the calibration procedure (Table 3). Overlaps of DMRT-plausible snow density ranges with the reference data at evaluation sites were similar

to Cambridge Bay, with slightly higher overlap at Alert and lower values for the other two sites. Similarly, the MAPEs of final snow density estimates, converted with H, at evaluation sites had slightly higher values than Cambridge Bay. Like Cambridge Bay, all sites displayed moderate positive correlations with the reference data, and biases had similar magnitude to Cambridge

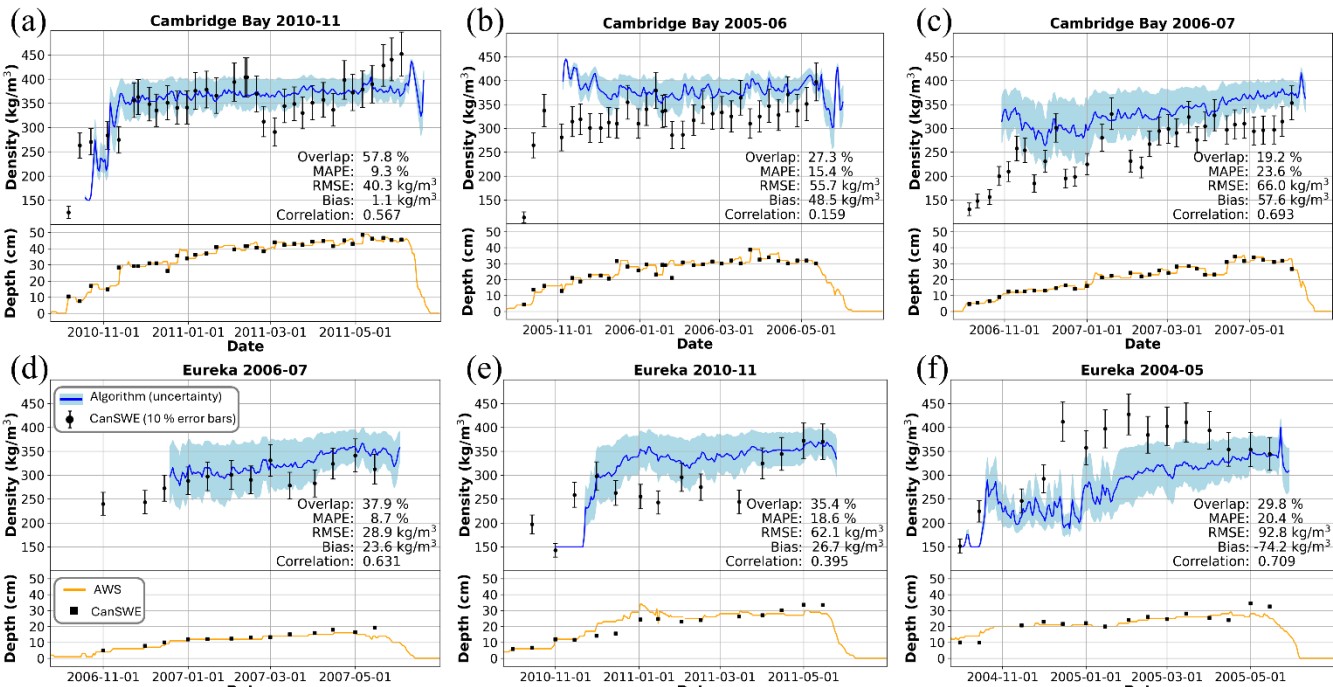

**Figure 3 - Examples of algorithm outputs at Cambridge Bay (a-c) and Eureka sites (d-f): a/d) better, b/e) moderate, c/f) worse performance. For each algorithm run, top panel shows algorithm snow density estimates and reference CanSWE snow density, and bottom panel shows AWS snow depth for algorithm forcing and reference CanSWE snow depth (legend for all panels shown in d).**



Bay (i.e. <10%). Like during calibration, estimation skill was not consistent over the study domain/period and the algorithm displayed varying levels of performance during different winter seasons throughout the study period. At Eureka, for example,

there were algorithm runs where estimates were very skilful (Fig 3d), and others where estimates did not agree with the reference data for parts of the winter season (Fig 3e&f).

## 6 Discussion

### 6.1 Seasonal Trends in Algorithm Performance

From the algorithm performance metrics in Figure 3 and Table 4, specifically those from the Eureka site, the estimation

skill improved over the course of a given winter season. In some winter seasons at Eureka the algorithm underestimated snow density early in the season or did not reflect early season variations in snow density (e.g. Fig 3b&c). However, algorithm estimates consistently improved over time and most algorithm estimates were close to the in situ references samples by the end of the algorithm run at the end of the season (i.e. within ±10%). To quantify this behaviour the reference dataset was partitioned into three seasonal sets – October-November-December (OND), January-February-March (JFM), and April-May-

June (AMJ) – and overlap, MAPE, and bias calculated for each set shown in Table 4. Algorithm estimates at Eureka in OND had low overlap with the in situ samples (17.2%) and were low biased (-51.2 kg/m$^3$; -18.2%) with relatively high MAPE (19.4%). Performance metrics improved in JFM for Eureka where overlap was more than double that of OND and MAPE and bias reduced. Performance metrics further improved in AMJ with >50% overlap and MAPE was within the uncertainty range of the reference samples. These results from Eureka suggest the algorithm configuration is less sensitive to early season snow

conditions and it could instead be better suited for retrievals later in the winter season. The behaviour of increasing algorithm estimation skill over the course of the winter season was apparent at the other sites but was less pronounced than at Eureka (Table 4).

**Table 4 – Seasonal performance metrics for algorithm snow density estimates relative to CanSWE, for October-November-December (OND), January-February-March (JFM), and April-May-June (AMJ).**

| Site | Overlap (%) | | | MAPE (%) | | | Bias (kg/m$^3$ [%]) | | |
|---|---|---|---|---|---|---|---|---|---|
| | OND | JFM | AMJ | OND | JFM | AMJ | OND | JFM | AMJ |
| Alert | 19.8 | 54.5 | 51.8 | 25.1 | 9.0 | 11.3 | 35.8 (11.9) | 24.4 (6.6) | 39.6 (10.4) |
| Eureka | 17.2 | 35.8 | 52.4 | 19.4 | 15.6 | 9.2 | -51.2 (-18.2) | -20.1 (-6.0) | 8.8 (2.6) |
| Resolute | 33.2 | 43.1 | 30.4 | 15.2 | 13.3 | 12.4 | 26.7 (7.7) | 29.1 (8.0) | 19.1 (4.9) |
| Cambridge Bay | 22.6 | 49.2 | 48.2 | 19.0 | 10.9 | 11.0 | 7.8 (2.6) | 3.0 (0.9) | 19.9 (5.8) |

The tendency of improved seasonal algorithm estimation skill did not appear to be related to seasonal differences in forcing data (i.e. situations with shallow snow depth or near freezing air temperature) and instead could be better explained by the algorithm configuration. The information available about tundra snow composition from field campaigns is biased towards the end of the winter season, typically occurring in March or April (e.g. Derksen et al., 2014; Meloche et al., 2022; Rees et al., 2014). Thus, it follows the configuration of the snowpack model would be most appropriate for the conditions towards the end





of the season, and the snowpack properties could be different early in the season (specifically the layer thickness ratio). Additionally, the estimates from the proposed algorithm are synoptic, representing general patterns over regional scales (25x25km$^2$), whereas the reference data from CanSWE covered more localized areas (snow courses along 150-300m transects). Snow distribution patterns in the high Arctic are terrain dependent and there can be considerable variability within a PM satellite footprint (Woo, 1998), so it was difficult to interpret the reference data in detail without specific information about where they were collected within the satellite observation scene.

## 6.2 Evaluation of Algorithm Configuration

Performance metrics for all sites (calibration and evaluation) were very similar suggesting the configuration of the electromagnetic snowpack model was appropriate for the high Arctic tundra environment, and the model calibration was not over fit to the Cambridge Bay site. The algorithm configuration appeared most appropriate towards the end of the season with considerable improvements at each site over the course of the winter season. However, there were winter seasons where algorithm estimates matched the reference data much better than others (Fig. 3&4). The radius of snow grains has a large effect on microwave emissions (Chang et al., 1982; Rango et al., 1979) and some of the year-to-year variability in algorithm agreement could be explained by the generalized calibration procedure for the snow grain growth model. The grain growth model parameters were the same for the whole study period, when there were likely different conditions between winter seasons and sites. For example, algorithm estimates for Cambridge Bay 2006-07 (Fig. 3c) displayed relative high correlation with the reference data (0.693) but with a large bias (53.9 kg/m$^3$; 17.5%) and high MAPE (23.6%); in this case, the grain radius estimates may have been too large and smaller values could bring snow density estimates closer to the reference samples. Overall, the algorithm configuration seemed suitable, given the similar results at the calibration and evaluation sites, but improvements could be made in how the microstructure was parameterized to better represent varying conditions and make algorithm performance more consistent.

There was one winter season at the Cambridge Bay site where the trajectory of algorithm estimates in the early winter season did not match the expected densification pattern. Intuitively snow should densify over time, yet during the 2005-06 run the algorithm estimated denser snow at the beginning of the winter season, with estimates decreasing over the early season rather than increasing (Fig 3b). That behaviour could be explained by the presence of water bodies around Cambridge Bay which are known to influence PM observations in the tundra environment when using the ΔTb modelling approach (Derksen et al., 2010). The generalized substrate representation in the electromagnetic model did not consider water bodies in the observation scene and could be modified to include water/lake ice to improve algorithm performance. However, the similar overall performance of the algorithm at Cambridge Bay (with many water bodies in the scene) and the evaluation sites (with virtually no water bodies in the scene) suggested the ΔTb approach was suitable.

The bulk snow density reference samples available in CanSWE did not allow for the densities of the individual snow layers to be calibrated, nor a thorough examination of the individual density values defined by the H parameter. Instead, density



measurements from Derksen et al.'s (2014) field campaign in April 2011 were used to provide some context about algorithm estimates for the two snow layers. Derksen et al. performed intensive snow surveys near Eureka (~50x50 km area) during that month and found the average wind slab layer and depth hoar layer densities to be 400 and 250 kg/m$^3$, respectively (combined

for a bulk density of 341 kg/m$^3$, very close to the CanSWE Eureka April snow density of 344 kg/m$^3$). Algorithm density estimates for the two snow layers derived with H [DMRT-plausible range] over the same period at Eureka (Fig 4b) were found to be comparable to those measured by Derksen et al. with wind slab and depth hoar densities of 380 kg/m$^3$ [320, 450] and 295 kg/m$^3$ [266, 320], respectively (combined for a bulk density of 352 kg/m$^3$ [320, 389]). So, the algorithm estimated bulk density is very close to that measured in the field at the regional-scale and the estimated wind slab layer density is also quite

similar (~5% lower) but the depth hoar density is overestimated by larger amount (~18% higher). While we cannot conclude from this limited sample size that the algorithm is perfect, the similarity of the algorithm estimates and layer densities to independent snow surveys suggest the parametrization of H was effective and that this approach could be expanded to estimate snow density across the tundra.

## 7 Conclusions and Future Work

A prototype algorithm was developed to estimate snow density in the tundra environment using PM remote sensing, given the challenges in estimating spatiotemporally varying snow density in that environment. An experiment was conducted to assess the algorithm's ability to estimate snow density at sites distributed in the Canadian tundra. Results from those sites demonstrate algorithm estimates of snow density provided information on snow density comparable to those collected with snow courses and appeared best suited for estimating snow density conditions later in the season. With more extensive forcing

data (e.g. snow depth estimates from global climate models) this algorithm could be applied over the tundra biome to provide snow density estimates at spatiotemporal scales that were not previously available.

The experimental design for this study was opportunistic due to the limited snow density data available for algorithm development and evaluation. CanSWE was the only readily available dataset which covered the required spatial and temporal domain for algorithm development but was limited to bulk estimates and, as result, the algorithm estimates for the two distinct

snow layers could not be sufficiently parameterized or evaluated. Tundra snow conditions are known to be driven by terrain types (Woo, 1998), and future algorithm development will focus on sites with distributed stratigraphic data to better quantify snow density conditions at the PM scale. By characterizing terrain variability at the regional scale, we hypothesise the DMRT-plausible range of snow densities for the PM scene could be disaggregated using high resolution active microwave data to provide information on stratigraphic heterogeneity and better estimate density values for the two distinct snow layers (to

replace the static H parameter).



*Code and data availability.* The retrieval algorithm and snow density estimates are still in the prototype phase and are not ready for distribution. Any inquiries can be submitted to the corresponding author.

*Author contributions.* JW developed the algorithm with guidance from RK. JW performed the analysis, produced the figures, and wrote the original manuscript draft, which was then edited by both authors.

*Competing interests.* The authors declare that they have no conflict of interest.

*Acknowledgements.* We acknowledge the support of the Natural Sciences and Engineering Research Council of Canada (NSERC), [funding reference number RGPIN-2023-04431].

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
