# Peer review of "A Prototype Passive Microwave Retrieval Algorithm for Tundra Snow Density"

_EGUsphere, 2024_

## Referee Comment (RC1)

**Revisions of: A Prototype Passive Microwave Retrieval Algorithm for Tundra Snow Density**
**By: Welch and Kelly**

Benoit Montpetit

**1   General Comments**

This paper tries to retrieve snow density from a two layer snowpack representation in the Canadian Arctic Tundra from passive microwave (PMW) remote sensing. To do so, they minimize a cost function between the measured brightness temperature delta ($\Delta$TB) at 37 and 19 GHz and the simulated $\Delta$TB using the Snow Microwave Radiative Transfer Model (SMRT). In order to simulate the $\Delta$TB, a number of assumptions on soil background properties, snow grain size and microstructure, snow temperature, and the snow depth hoar thickness ratio are made. Finally, many of these assumptions need to be empirically calibrated in order to fit measured TBs.

The manuscript is well written and figures greatly support the text and help visualize the methodology and the results. Though I recognize and appreciate the amount of work that has been put into this study, I question some of these assumptions and would highly recommend revising the strategy used to retrieve and validate the tundra density profiles. In its current state, it is impossible to determine if the retrieval is representative of the reality or if the calibrations are simply compensating for errors coming from the assumptions made, thus providing invalid density values. Below is a list of assumptions I question and, if possible, suggest potential alternatives.

I also suggest looking at the work of Woolley et al. (2024); Meloche et al. (2022), where they present physical and statistical representations of snowpack properties for tundra snowpacks at scales similar to PMW. I would also look at the methodology of Picard et al. (2022) where they retrieved snowpack properties from passive microwave to analyze the sensitivity of PMW observations to one parameter in the snowpack (e.g. liquid water). A similar approach could be done here to test the PMW sensitivity to density.

**1.1 Using weather station data to estimate TB with SMRT**

Using AWS data (point scale) to estimate snow/soil conditions is not representative of the PMW scale (25 km). This can induce major errors in the retrieval process. I highly suggest looking at reanalysis datasets like ERA-5 to estimate the snow/soil properties. Its scale is much better suited for PMW data comparison. Also, it is possible to get more than one grid-cell of the reanalysis data within the PMW pixel. Using such datasets also supports further comments below.

Meloche et al. (2022) has shown that a high coefficient of variation on snow depth, i.e. using a distributed range of snow
depth values, is better suited for SWE retrievals at the 25km scale. This means that using a single value for snow depth and depth hoar fraction will induce errors on the retrieved parameters since the optimization process optimizes for inaccuracies in the assumed parameters. The also showed in this study that using a static depth hoar fraction, both spatially and temporally, is not representative of what is detected by PMW at the 25 km resolution.

Another variable that could be considered while using re-analysis data is the lake fraction effect which is not considered in
this study. The authors mention topography and forest cover, which needs to be minimized but they do not mention anything about lake fraction.

Using a distributed approach using: 1) a more representative source of data, and 2) comparing it with AWS data would be necessary to confirm that the retrieved densities are valid and that these retrieved values do not compensate for errors in initial assumptions.

**1.2 Choice of DMRT with non-sticky spheres**

Vargel et al. (2020); Royer et al. (2017); Roy et al. (2016); Liang et al. (2008) have shown that choosing the non-sticky sphere version of DMRT is not suitable to simulate TBs. The grain size need to be compensated by a scaling factor due to the stickiness of spheres (Roy et al., 2016). Vargel et al. (2020) showed that results converge towards a stickiness of $\tau$=0.1 which confirms the work of Liang et al. (2008).

That said, more evidence has been gathered in these studies that DMRT-QCA is not the best suited theory to simulate the scattering of depth hoar layers (Vargel et al., 2020). This is why the improved born approximation is now commonly used to simulate microwave signals, both active and passive (Montpetit et al., 2024; Sandells et al., 2022; Vargel et al., 2020; King et al., 2018). Only when densities are above 400 kg·m$^{-3}$ should the strong contrast expansion theory be considered (Meloche et al., 2024).

Keeping the validity limit of $450\,\text{kg}\cdot\text{m}^{-3}$ thus needs to be justified and the scattering theory for this study has to be modified accordingly. That said, with SMRT, different scattering theories can be applied to different layers of the same snowpack (Picard et al., 2018) and these theories can evolve throughout the season as the snowpack evolves.

**1.3  Choosing the (Kelly et al., 2003) grain growth to simulate TBs**

This model needs to be validated against observations of optical radius grain sizes. At the time of this growth model, this
parameter was very difficult to quantify in the field. Since then, many instruments have been developed and should be used against field measurements. Many datasets of snow grain size were acquired and reported in the literature for the studied sites. I highly suggest comparing this grain growth model to these datasets to validate it. Otherwise, I would look at mean values and include an uncertainty to it to retrieve density, since grain size is one of the most, if not the most, important parameter to simulate PMW emissivity. In the current methodology, I would leave this parameter free and optimize it with density in order to
assess the uncertainty on both parameters with a sensitivity analysis. By setting the grain size value, it is difficult to determine if the retrieval method is compensating for poor grain size estimate. This could explain higher errors in the early season.

**1.4  Using 2m air temperature to estimate depth hoar layer temperature**

Knowing that snow in the tundra has a high temperature gradient between the snow-air and soil-air interfaces, and that the PMW signal is sensitive to layer temperature, it is not representative to estimate the temperature of both layers with the air
temperature.

I suggest using soil surface temperature as a proxy for the depth hoar layer temperature. This might be available from AWS, and is definitely available in reanalysis datasets.

**1.5  Using the (Dobson et al., 1985) model to estimate the frozen ground permittivity**

Zhang et al. (2010) mentions that the permittivity values calculated by the Dobson et al. (1985) model are too high and suggests
a different model to estimate frozen soil permittivity. Montpetit et al. (2018) showed that both permittivity and roughness are important background properties to estimate PMW emissivity. Meloche et al. (2021) has shown that using the Wegmüller and Mätzler (1999) roughness model with the Zhang et al. (2010) permittivity model with the average root-mean-square height of 1.65 cm gave the best results.

It is highly recommended that the results of Meloche et al. (2021) be included in this study to properly simulate TBs.

 **1.6 Using a static depth hoar ratio to simulate seasonally evolving tundra snowpacks**

See above comments from the work of Meloche et al. (2022). The depth hoar layer has a significant impact on PMW emissivity. Depth hoar fraction is highly variable both spatially and temporally. This alone could explain why the retrieved densities are closer at the end of the seasons.

Including information such as what is described in Woolley et al. (2024), i.e. a more dynamic depth hoar fraction, could improve the temporal accuracy per site and improve the inter-site comparison where different mean depth hoar fraction are most likely representative of the four sites analyzed in this study.

**1.7 Not considering atmospheric contributions to the simulated TBs**

Sandells et al. (2024) shows the importance of considering atmospheric conditions in TB simulations. Though the study was conducted at higher frequencies than the ones used in this study, similar conclusions were found in the following studies (Montpetit et al., 2013; Roy et al., 2013; Meloche et al., 2022). Vargel et al. (2020) also showed the importance of considering atmospheric contributions. They even showed that the contributions can be very different at 19 and 37 GHz which will have a considerable impact on the simulated $\Delta$TB. GlobSnow products also consider atmospheric conditions to retrieve SWE (Yang et al., 2024; Zschenderlein et al., 2023).

It is thus crucial that this be considered and the methodology used by previous studies (described in Vargel et al. (2020)) is a good method to consider atmospheric contributions using reanalysis datasets.

**1.8 Using a "brute-force" method to optimize the cost function**

The proposed method of reducing the cost function between $\Delta$TBs is very reliant on the above assumptions which need to be justified and validated.

There are more robust methods that have been implemented in Picard et al. (2022); Pan et al. (2017); Meloche et al. (2022) to retrieve snowpack properties from PMW. The a priori knowledge provided by the validated assumptions presented here could prove more suitable to retrieve profiled density.

**1.9 Validating retrieved densities with CanSWE**

The CanSWE dataset is an excellent source to compare and validate the retrieved densities from this study. That said, it is impossible to identify where the errors originate from precisely since what is compared is bulk density and the proposed assumptions are fixed. For example, a fixed depth hoar fraction does not allow to estimate the sensitivity of the retrieval method to the thickness of both layers.

In order to better assess the efficiency and accuracy of the retrieval method, a robust sensitivity analysis has to be conducted to properly identify the sources of errors and have more robust and plausible explanations on the discrepancies for certain sites and times.

**100 References**

Dobson, M., Ulaby, F., Hallikainen, M., and El-Rayes, M.: Microwave Dielectric Behavior of Wet Soil-Part II: Dielectric Mixing Models, IEEE Transactions on Geoscience and Remote Sensing, GE-23, 35–46, https://doi.org/10.1109/TGRS.1985.289498, 1985.

Kelly, R., Chang, A., Tsang, L., and Foster, J.: A Prototype AMSR-E Global Snow Area and Snow Depth Algorithm, IEEE Transactions on Geoscience and Remote Sensing, 41, 230–242, https://doi.org/10.1109/TGRS.2003.809118, 2003.

King, J., Derksen, C., Toose, P., Langlois, A., Larsen, C., Lemmetyinen, J., Marsh, P., Montpetit, B., Roy, A., Rutter, N., and Sturm, M.: The Influence of Snow Microstructure on Dual-Frequency Radar Measurements in a Tundra Environment, Remote Sensing of Environment, 215, 242–254, https://doi.org/10.1016/j.rse.2018.05.028, 2018.

Liang, D., Xu, X., Tsang, L., Andreadis, K., and Josberger, E.: The Effects of Layers in Dry Snow on Its Passive Microwave Emissions Using Dense Media Radiative Transfer Theory Based on the Quasicrystalline Approximation (QCA/DMRT), IEEE Transactions on Geoscience 110 and Remote Sensing, 46, 3663–3671, https://doi.org/10.1109/TGRS.2008.922143, 2008.

Meloche, J., Royer, A., Langlois, A., Rutter, N., and Sasseville, V.: Improvement of Microwave Emissivity Parameterization of Frozen Arctic Soils Using Roughness Measurements Derived from Photogrammetry, International Journal of Digital Earth, 14, 1380–1396, https://doi.org/10.1080/17538947.2020.1836049, 2021.

Meloche, J., Langlois, A., Rutter, N., Royer, A., King, J., Walker, B., Marsh, P., and Wilcox, E.: Characterizing Tundra Snow Sub-Pixel 115 Variability to Improve Brightness Temperature Estimation in Satellite SWE Retrievals, Cryosphere, 16, 87–101, https://doi.org/10.5194/tc-16-87-2022, 2022.

Meloche, J., Royer, A., Roy, A., Langlois, A., and Picard, G.: Improvement of Polar Snow Microwave Brightness Temperature Simulations for Dense Wind Slab and Large Grain, IEEE Transactions on Geoscience and Remote Sensing, pp. 1–1, https://doi.org/10.1109/TGRS.2024.3428394, 2024.

Montpetit, B., Royer, A., Roy, A., Langlois, A., and Derksen, C.: Snow Microwave Emission Modeling of Ice Lenses within a Snowpack Using the Microwave Emission Model for Layered Snowpacks, IEEE Transactions on Geoscience and Remote Sensing, 51, 4705–4717, https://doi.org/10.1109/TGRS.2013.2250509, 2013.

Montpetit, B., Royer, A., Roy, A., and Langlois, A.: In-Situ Passive Microwave Emission Model Parameterization of Sub-Arctic Frozen Organic Soils, Remote Sensing of Environment, 205, 112–118, https://doi.org/10.1016/j.rse.2017.10.033, 2018.

Montpetit, B., King, J., Meloche, J., Derksen, C., Siqueira, P., Adam, J. M., Toose, P., Brady, M., Wendleder, A., Vionnet, V., and Leroux, N. R.: Retrieval of Snow and Soil Properties for Forward Radiative Transfer Modeling of Airborne Ku-band SAR to Estimate Snow Water Equivalent: The Trail Valley Creek 2018/19 Snow Experiment, The Cryosphere, 18, 3857–3874, https://doi.org/10.5194/tc-18-3857-2024, 2024.

Pan, J., Durand, M. T., Vander Jagt, B. J., and Liu, D.: Application of a Markov Chain Monte Carlo Algorithm for Snow Water Equivalent Re-130 trieval from Passive Microwave Measurements, Remote Sensing of Environment, 192, 150–165, https://doi.org/10.1016/j.rse.2017.02.006, 2017.

Picard, G., Sandells, M., and Löwe, H.: SMRT: An Active-Passive Microwave Radiative Transfer Model for Snow with Multiple Microstructure and Scattering Formulations (v1.0), Geoscientific Model Development, 11, 2763–2788, https://doi.org/10.5194/gmd-11-2763-2018, 2018.

Picard, G., Leduc-Leballeur, M., Banwell, A. F., Brucker, L., and Macelloni, G.: The Sensitivity of Satellite Microwave Observations to Liquid Water in the Antarctic Snowpack, The Cryosphere, 16, 5061–5083, https://doi.org/10.5194/tc-16-5061-2022, 2022.

Roy, A., Picard, G., Royer, A., Montpetit, B., Dupont, F., Langlois, A., Derksen, C., and Champollion, N.: Brightness Temperature Simulations of the Canadian Seasonal Snowpack Driven by Measurements of the Snow Specific Surface Area, IEEE Transactions on Geoscience and Remote Sensing, 51, 4692–4704, https://doi.org/10.1109/TGRS.2012.2235842, 2013.

Roy, A., Royer, A., St-Jean-Rondeau, O., Montpetit, B., Picard, G., Mavrovic, A., Marchand, N., and Langlois, A.: Microwave Snow Emission Modeling Uncertainties in Boreal and Subarctic Environments, Cryosphere, 10, 623–638, https://doi.org/10.5194/tc-10-623-2016, 2016.

Royer, A., Roy, A., Montpetit, B., Saint-Jean-Rondeau, O., Picard, G., Brucker, L., and Langlois, A.: Comparison of Commonly-Used Microwave Radiative Transfer Models for Snow Remote Sensing, Remote Sensing of Environment, 190, 247–259, https://doi.org/10.1016/j.rse.2016.12.020, 2017.

Sandells, M., Löwe, H., Picard, G., Dumont, M., Essery, R., Floury, N., Kontu, A., Lemmetyinen, J., Maslanka, W., Morin, S., Wiesmann, A., and Mätzler, C.: X-Ray Tomography-Based Microstructure Representation in the Snow Microwave Radiative Transfer Model, IEEE Transactions on Geoscience and Remote Sensing, 60, https://doi.org/10.1109/TGRS.2021.3086412, 2022.

Sandells, M., Rutter, N., Wivell, K., Essery, R., Fox, S., Harlow, C., Picard, G., Roy, A., Royer, A., and Toose, P.: Simulation of Arctic Snow

Microwave Emission in Surface-Sensitive Atmosphere Channels, Cryosphere, 18, 3971–3990, https://doi.org/10.5194/tc-18-3971-2024, 2024.

Vargel, C., Royer, A., St-Jean-Rondeau, O., Picard, G., Roy, A., Sasseville, V., and Langlois, A.: Arctic and Subarctic Snow Microstructure Analysis for Microwave Brightness Temperature Simulations, Remote Sensing of Environment, 242, https://doi.org/10.1016/j.rse.2020.111754, 2020.

Wegmüller, U. and Mätzler, C.: Rough Bare Soil Reflectivity Model, IEEE Transactions on Geoscience and Remote Sensing, 37, 1391–1395, https://doi.org/10.1109/36.763303, 1999.

Woolley, G. J., Rutter, N., Wake, L., Vionnet, V., Derksen, C., Essery, R., Marsh, P., Tutton, R., Walker, B., Lafaysse, M., and Pritchard, D.: Multi-Physics Ensemble Modelling of Arctic Tundra Snowpack Properties, EGUsphere, pp. 1–38, https://doi.org/10.5194/egusphere-2024-1237, 2024.

Yang, J., Jiang, L., Lemmetyinen, J., and Luojus, K.: A New Method to Simulate the Microwave Effective Snow Grain Size in the Northern Hemisphere without Using Snow Depth Priors, IEEE Journal of Selected Topics in Applied Earth Observations and Remote Sensing, pp. 1–18, https://doi.org/10.1109/JSTARS.2024.3441817, 2024.

Zhang, L., Zhao, T., Jiang, L., and Zhao, S.: Estimate of Phase Transition Water Content in Freeze-Thaw Process Using Microwave Radiometer, IEEE Transactions on Geoscience and Remote Sensing, 48, 4248–4255, https://doi.org/10.1109/TGRS.2010.2051158, 2010.

Zschenderlein, L., Luojus, K., Takala, M., Venäläinen, P., and Pulliainen, J.: Evaluation of Passive Microwave Dry Snow Detection Algorithms and Application to SWE Retrieval during Seasonal Snow Accumulation, Remote Sensing of Environment, 288, https://doi.org/10.1016/j.rse.2023.113476, 2023.

---

## Author Comment (AC1)

**Review Response to Michael Durand**

Dr. Durand,

Thank you very much for your thoughtful review of our manuscript. Your suggestions are very helpful and we will work to incorporate them into the revised manuscript. Specifically, the suggestions regarding including a sensitivity test are important and we will design experiments to demonstrate the effect of snow density on brightness temperature and support the algorithm parameterization. In the future, we plan to apply the algorithm across the pan-Arctic and your suggestions will help to justify the methodology. Below are responses to the individual review comments.

1. Sensitivity tests

This is a very good point and we agree that a description of the influence of snow density on brightness temperature and sensitivity tests should be included in the revised manuscript – we will incorporate your ideas (i.e. varying density with constant snow depth and constant SWE) and have a couple ideas of our own for potential scenarios to include (e.g. effect of layer thicknesses). Thinking about this suggestion inspired a new way to handle, and justify, the layer thickness parameters in the model.

2. Units on Figure 2.

Agreed, and we will modify Figure 2 accordingly to provide more context on the sensitivity to the layer density parameters in the modelled output.

3. Brightness temperature in Figure 3.

This information was not included in Figure 3 in an effort to simplify the figures and make them easier to read – although, we recognized it is very important and will include those data in the revised manuscript.

4. Equation relating H to the layer densities

Upon review, our explanation of the heterogeneity parameter (H) was too brief and needs more explanation. We will elaborate on how the H parameter works and include the formulas to convert the plausible range to estimates for the specific layers using H.

5. Fixed depth hoar ratio

Regarding the two papers you cited: the King et al (2015) experiment was located in the Hudson Bay Lowlands which is a sub-Arctic transitional environment and not an Arctic snowpack. This study focuses on the high Arctic which is characterized by a relatively thinner depth hoar. Your comment regarding the Zhu et al (2018) study is interesting. But Zhu uses active microwave observations for which volume scattering is the dominant mechanism considered in both layers (after isolating it from surface and

background scattering components). For passive microwave observations, volume scattering is the primary mechanism for the depth hoar layer and non-scattering emission contributions originate from the wind slab (Sturm et al., 1993). This second point (different contribution sources from the layers to passive microwave brightness temperature) has made us reconsider the usefulness of a static thickness ratio and instead are considering introducing a maximum depth hoar thickness (dependent on site characteristics, e.g. predominant vegetation types), which we think can be supported by sensitivity tests.

6. Equation for prediction of bulk density from H and layer ratio of depths

Similar to point 4, we think this is a good idea and will include an equation for deriving the final bulk density from the layer estimates derived with H.

7.  Calibration of H parameter

The calibration of H would likely make more sense if the effect of H was more explicitly described in the manuscript (including equations, as per point 4 above). It seems H might have been confused with the layer thickness ratio, instead it is meant to quantify the difference in densities between the layers – ranging from the most conservative estimate where the layer densities are identical (i.e. minimum heterogeneity) to the situation with the most distinct contrast between layers (i.e. maximum heterogeneity). Again, this will be explained more thoroughly in the revised manuscript and visual example included in Figure 2.

**References**

King, J., Kelly, R., Kasurak, A., Duguay, C., Gunn, G., Rutter, N., et al. (2015). Spatio-temporal influence of tundra snow properties on Ku-band (17.2 GHz) backscatter. Journal of Glaciology, 61(226), 267-279(13). https://doi.org/10.3189/2015jog14j020

Sturm, M., Grenfell, T. C., & Perovich, D. K. (1993). Passive microwave measurements of tundra and taiga snow covers in Alaska, USA. *Annals of Glaciology*, *17*, 125–130. https://doi.org/10.3189/s0260305500012714

Zhu, J., Tan, S., King, J., Derksen, C., Lemmetyinen, J., & Tsang, L. (2018). Forward and Inverse Radar Modeling of Terrestrial Snow Using SnowSAR Data. IEEE Transactions on Geoscience and Remote Sensing, 56(12), 7122–7132. https://doi.org/10.1109/tgrs.2018.2848642

---

## Author Comment (AC2)

**Review Response to Bennoit Montpetit**

Dr. Montpetit,

Thank you for your thorough review of your manuscript, specifically regarding the parameterization of the snowpack model. We believe a distinction should be made between small scale field studies and hemispheric scale retrieval algorithms; the latter require substantial generalization in order to be applied broadly, which is the ultimate goal. The proposed algorithm is in a prototype phase to demonstrate its applicability for the estimation of snow density conditions from passive microwave remote sensing. In the future, we plan to build upon the prototype algorithm to better characterize the snowpack model across large areas and facilitate pan-Arctic snow density retrievals. Below are responses to the specific review comments.

1.1 Using weather station data to estimate TB with SMRT

We recognize the limitations of AWS data for capturing spatial variability, specifically with respect to snow depth. However, we do not think reanalysis data would necessarily be appropriate here - Cao et al. (2020) showed that ERA-5 demonstrates high biased snow depth in high latitudes. We limited our input datasets to seasons where the AWS SD measurements were similar to those measured over snow courses (i.e. CanSWE) – so they should be somewhat similar to spatially averaged depth (we can probably state this more clearly in the manuscript). Additionally, we are looking to revise the layer thickness ratio which will reduce the model sensitivity to the AWS snow depth (explained further under 1.6).

Regarding ground temperature, operational AWS do not measure ground temperature in the Arctic. The confidence in ground temperature estimates from reanalysis data is limited during the cold season (Herrington et al., 2024), and ERA-5 exhibits high biased estimates of ground temperature in high latitudes (Cao et al., 2020). The absolute accuracy of ground temperatures is less important when using a frequency difference approach – we believe our parameterization of ground temperatures relative to snow temperature is in the right ballpark to characterize the effect of thermal emission from the snowpack (specifically from the wind slab layer).

In terms of lake fraction, there should be minimal effect for the 3 northern most sites, though it is an important consideration for Cambridge Bay – we will mention this in the study site section and include lake fraction estimates for each site. Given the high latitude of Cambridge Bay, we expect the influence of lake fraction to be minimal later in the season. In the future, we will look into handling lake ice fraction more explicitly.

1.2 Choice of DMRT with non-sticky spheres

Your point about DMRT-QCA is valid and we will look into using IBA based on Meloche et al. (2024). However, we are not convinced that introducing stickiness into the microstructure representation is the best way forward, because it is an unmeasurable property and is essentially a tuning parameter. Roy et al. (2016) used a non-sticky case with a scaling factor (from Roy et al., 2013) to convert from optical to effective grain size. Further, Roy et al. (2013) concluded "even if the stickiness seems to be a

pertinent physical explanation, in practice, its introduction poses difficulties because it is not a measurable quantity for natural snow and its optimization is not unequivocal, even in the simplified case that we considered here by using a constant value for the entire snowpack". Vargel et al. (2020) use stickiness to compensate for optical grain size (rather than effective grain size) – whereas we optimize for effective grain size directly removing the need for a stickiness compensation factor.

1.3 Choosing the (Kelly et al., 2003) grain growth to simulate TBs

Sturm et al.'s (1997) kinetic grain growth model is based on empirical observations. While that model estimates physical grain size (rather than effective radius), we argue it should be representative of the relative change in grain radius. We plan to modify the grain size optimization process to focus more on end of season conditions and compare them to values reported in the literature. Further, recent works (e.g. Wooley et al., 2024; Meloche et al., 2022) have shown inter-season variability in grain size (SSA) is low relative to changes in density conditions – therefore, we believe it is appropriate to use an optimized grain growth model to retrieve density parameters.

1.4 Using 2m air temperature to estimate depth hoar layer temperature

We agree that the 2m air temperature is not representative of depth hoar temperature, reflecting an oversimplification of our model, and will modify this parameter. We plan to replace the homogenous snowpack temperature with a linear temperature gradient. However, the temperature of the depth hoar layer will have a very minor impact on the simulated brightness temperatures (i.e. <1K), as emission contribution from that layer is much lower than that of the wind slab layer.

1.5  Using the (Dobson et al., 1985) model to estimate the frozen ground permittivity

Thank you for the suggestion, we will incorporate the findings from Meloche et al (2021). However, the effects of the substrate should be minimal using a frequency difference modelling approach and we do not think it will change the results by a considerable amount.

1.6 Using a static depth hoar ratio to simulate seasonally evolving tundra snowpacks

It is possible that we could incorporate the relationship between snow depth and depth hoar ratio from Meloche et al. (2022), however that work does not consider temporal evolution and is restricted to end of season conditions. Woolley et al. (2024) relies on stratigraphy from snow pits to properly segment their modelled density profiles for evaluation – while we acknowledge it is important to consider variable layer thickness ratios, that paper does not involve the prediction of layer ratios but instead relies on in situ measurements to interpret their modelled results. Currently, there is insufficient data available to properly parameterize the seasonal evolution of tundra snow density profiles, and we mention that is likely the cause for improved density retrievals towards the end of the season (on line 249). Moving forward, we are looking into including a maximum depth hoar thickness for each site

(considering site characteristics like vegetation; Domine et al., 2016) , rather than a fixed ratio based on snow depth to reduce the sensitivity to the AWS snow depth measurements.

1.7 Not considering atmospheric contributions to the simulated TBs

We recognize that atmospheric contributions should be considered and reflects an oversimplification of our modeling approach. That being said, we do not think that a complex representation is strictly necessary in this case. You mention GlobSnow's handling of atmospheric corrections, yet the two references you provided do not mention any specifics about atmospheric contributions. Instead, GlobSnow uses an empirical atmospheric model that uses static parameters over space and time. We plan to incorporate the atmospheric model used in GlobSnow, parameterized with AWS data (rather than using a static air temperature of -5C used in GlobSnow).

1.8 Using a "brute-force" method to optimize the cost function

We are unsure of what you mean by "brute-force" in our optimization procedure – it does use assumed parameters but that does not necessarily make it brute-force. On the other hand, the MCMC methods which you referenced could be more accurately characterized as "brute-force" methods, by essentially trying every possible parameter combination. We do not think the MCMC approach is necessary here and would be difficult to apply broadly (which is our ultimate goal) because so many of the required parameters (i.e. distributions of grain sizes) are not well known.

The reviewer commented about the assumptions made in the model and the need to calibrate "many" of those assumptions. We would like to clarify that only the effective grain size parameter was calibrated (along with the H parameter) and the other parameters were forced by AWS observations or informed from the available literature.

1.9 Validating retrieved densities with CanSWE

We address the limitations of the CanSWE dataset in the manuscript (line 303), and we acknowledge that its bulk nature is not ideal to evaluate the two layered model. However, no other dataset covers the necessary spatiotemporal scales to evaluate the model on large scales. That being said, we are in the process of acquiring stratigraphic data for the Eureka site from Derksen et al. (2014) to better evaluate the estimated layer densities.

**References**

Cao, B., Gruber, S., Zheng, D., and Li, X.: The ERA5-Land soil temperature bias in permafrost regions, Cryosphere, 14, 2581–2595, https://doi.org/10.5194/tc-14-2581-2020, 2020.

Derksen, C., Lemmetyinen, J., Toose, P., Silis, A., Pulliainen, J., and Sturm, M.: Physical properties of arctic versus subarctic snow: Implications for high latitude passive microwave snow water equivalent retrievals, J Geophys Res, 119, 7254–7270, https://doi.org/10.1002/2013JD021264, 2014.

Dobson, M., Ulaby, F., Hallikainen, M., and El-Rayes, M.: Microwave Dielectric Behavior of Wet Soil-Part II: Dielectric Mixing Models, IEEE Transactions on Geoscience and Remote Sensing, GE-23, 35–46, https://doi.org/10.1109/TGRS.1985.289498, 1985.

Domine, F., Barrere, M., and Morin, S.: The growth of shrubs on high Arctic tundra at Bylot Island: Impact on snow physical properties and permafrost thermal regime, Biogeosciences, 13, 6471–6486, https://doi.org/10.5194/bg-13-6471-2016, 2016.

Herrington, T. C., Fletcher, C. G., and Kropp, H.: Validation of pan-Arctic soil temperatures in modern reanalysis and data assimilation systems, Cryosphere, 18, 1835–1861, https://doi.org/10.5194/tc-18-1835-2024, 2024.

Meloche, J., Royer, A., Langlois, A., Rutter, N., and Sasseville, V.: Improvement of Microwave Emissivity Parameterization of Frozen Arctic Soils Using Roughness Measurements Derived from Photogrammetry, International Journal of Digital Earth, 14, 1380–1396, https://doi.org/10.1080/17538947.2020.1836049, 2021.

Meloche, J., Langlois, A., Rutter, N., Royer, A., King, J., Walker, B., Marsh, P., and Wilcox, E.: Characterizing Tundra Snow Sub-Pixel Variability to Improve Brightness Temperature Estimation in Satellite SWE Retrievals, Cryosphere, 16, 87–101, https://doi.org/10.5194/tc16-87-2022, 2022

Meloche, J., Royer, A., Roy, A., Langlois, A., and Picard, G.: Improvement of Polar Snow Microwave Brightness Temperature Simulations for Dense Wind Slab and Large Grain, IEEE Transactions on Geoscience and Remote Sensing, pp. 1–1, https://doi.org/10.1109/TGRS.2024.3428394, 2024.

Roy, A., Picard, G., Royer, A., Montpetit, B., Dupont, F., Langlois, A., Derksen, C., and Champollion, N.: Brightness temperature simulations of the Canadian seasonal snowpack driven by measurements of the snow specific surface area, IEEE Transactions on Geoscience and Remote Sensing, 51, 4692–4704, https://doi.org/10.1109/TGRS.2012.2235842, 2013.

Roy, A., Royer, A., St-Jean-Rondeau, O., Montpetit, B., Picard, G., Mavrovic, A., Marchand, N., and Langlois, A.: Microwave snow emission modeling uncertainties in boreal and subarctic environments, Cryosphere, 10, 623–638, https://doi.org/10.5194/tc-10-623-2016, 2016.

Sturm, M. and Benson, C. S.: Vapor transport, grain growth and depth-hoar development in the subarctic snow, Journal of Glaciology, 43, https://doi.org/10.3189/S0022143000002793, 1997.

Vargel, C., Royer, A., St-Jean-Rondeau, O., Picard, G., Roy, A., Sasseville, V., and Langlois, A.: Arctic and subarctic snow microstructure analysis for microwave brightness temperature simulations, Remote Sens Environ, 242, https://doi.org/10.1016/j.rse.2020.111754, 2020.

Woolley, G. J., Rutter, N., Wake, L., Vionnet, V., Derksen, C., Essery, R., Marsh, P., Tutton, R., Walker, B., Lafaysse, M., and Pritchard, D.: Multi-Physics Ensemble Modelling of Arctic Tundra Snowpack Properties, EGUsphere, pp. 1–38, https://doi.org/10.5194/egusphere2024-1237, 2024.

---

## Author Response (AR1)

**Author's Response - Major Revision #1**

**In response to Dr. Benoit Montpetit:**

**1.1 Using weather station data to estimate TB with SMRT**

- Statistical metrics for agreement between AWS and CanSWE included in Table 2
- Modified handling of ground temperature using a simple model based on AWS air temperature described in section 4.4.3, comparison with record temperatures (at other Arctic sites; Domine et al., 2018) included in Appendix A.
- Lake fractions added to Table 1 (along with vegetation characteristics)

**1.2 Non-sticky sticky hard spheres - Choice of DMRT with non-sticky spheres**

Electromagnetic model switched to IBA with microwave grain size microstructure (described in Section 4.1)

**1.3 Choosing the (Kelly et al., 2003) grain growth to simulate TBs**

- Switched to SSA given microstructure change described above in 1.2
- SSA parameterization described in Section 4.4.4.

**1.4 Using 2m air temperature to estimate depth hoar layer temperature**

• Snow temperatures parameterized with linear gradient (Section 4.4.3).

**1.5 Using the (Dobson et al., 1985) model to estimate the frozen ground permittivity**

• Parameterized substrate following Meloche et al. (2021).

**1.6 Using a static depth hoar ratio to simulate seasonally evolving tundra snowpacks 1.7 Including basic atmospheric contribution at these frequencies is sensible – please do so**

- Depth hoar parameterized as explicit thickness (rather than fraction)
- Depth hoar thickness left free in calibration procedure
- Comparison of dynamic depth hoar thickness, fixed thickness, and fixed fraction included in Section 4.3
- Depth Hoar Index proposed to estimate end-of-season depth hoar thickness, and expand over early season (Section 4.4.1) similar to Lievens et al.'s (2019) Snow Index for c-band SAR

**1.7 Not considering atmospheric contributions to the simulated TBs**

• Simple atmospheric contributions considered following approach used in GlobSnow (i.e. Pulliainen and Grandeil, 1999)

**1.8 Using a "brute-force" method to optimize the cost function**

• No changes made

**1.9 Validating retrieved densities with CanSWE**

 Acquired stratigraphic data from Saberi et al. (2017), used to evaluate the algorithm calibrate to bulk density samples from CanSWE (Section 5.2)

**In response to Dr. Michael Durand:**

**1. Sensitivity tests**

- Sensitivity tests included to describe effect of snow density on simulated brightness temperature (considering variations in SSA, polydispersity, and layer thickness) and the effect of wind compaction (Section 4.2)
- An additional sensitivity test was drafted for the effect of layer temperature but was not included in the revised manuscript (included at the end of this document) instead a single line was included on line 195

**2. Modifications to Figure 2 (now Figure 6)**

- Added units to contour lines in Figure 6a
- Added second panel (Figure 6b) showing mapping to bulk density

**3. Modifications to Figure 3 (now Figure 7)**

• Brightness temperature gradient included in Figure 7, along with other forcing data (snow depth, depth hoar thickness, air temperature, and simulated ground temperature)

**4. Description of H Parameter**

• More in depth description of H included in Section 4.3, including equations (see point 6 below)

**5. Static Depth Hoar Fraction**

See 1.6 above

**6. Equations for Final Density Estimates from H**

• Added equations showing estimation of individual layer densities with H (Eqs. 2&3) and conversion to bulk density (Eq. 4) in Section 4.3

**7. Calibration Questions**

- Calibration results discussed and compared for all sites (Section 6.1)
- Example solutions at various H levels included in Figure 6

**Additional Changes**

- Changed radiometry data to Enhanced-Resolution Passive Microwave Daily Brightness Temperature Version 2 dataset (Brodzik et al., 2024), from 25 km L2a product
  - Allowed EASE grid cells to be closer to AWS/CanSWE site
  - Better discernment between Eureka snow course and snow survey data (Saberi et al., 2017)
  - See updated Figure 1
- Included brief description of other passive microwave snow density retrieval algorithms on line 50
- Identified potential stages of snowpack evolution to support temporal parameterization (Section 4.4)
- Included all algorithm simulations in Appendix B
- Editing for grammar/readability throughout
- Draft sensitivity test for layer temperatures (not currently included in manuscript, see point 1 above):

Experiment 4 showed thermal emission from the wind slab is an important consideration for our model, so Experiment 5 was designed to demonstrate the effect of varying temperatures of the various media in the electromagnetic model (Fig 5). Three situations were considered where air or substrate temperature were modified by  $\pm 10$  K, while the other kept constant, or where all temperatures were modified together by  $\pm 10$  K. In all cases there was a linear temperature gradient in the snowpack and changes to air temperature would be

reflected in snow temperature. Absolute temperature of the various media has little effect on simulated  $\Delta Tb$  (

Figure 5 – Simulated microwave emission of two-layer snowpack with modified layer temperatures.

**References**

- Brodzik, M. J., Long, D. G., and Hardman, M. A.: Calibrated Passive Microwave Daily EASE-Grid 2.0 Brightness Temperature ESDR (CETB) Algorithm Theoretical Basis Document Version 2.1, https://doi.org/10.5281/zenodo.11626219, 2024.
- Domine, F., Belke-Brea, M., Sarrazin, D., Arnaud, L., Barrere, M., and Poirier, M.: Soil moisture, wind speed and depth hoar formation in the Arctic snowpack, Journal of Glaciology, 64, 990–1002, https://doi.org/10.1017/jog.2018.89, 2018.
- Gelinas, A., Filali, B., Langlois, A., Kelly, R., Mavrovic, A., Demontoux, F., & Roy, A.: New wideband large aperture open-ended coaxial microwave probe for soil dielectric characterization. IEEE Transactions on Geoscience and Remote Sensing, 63, https://doi.org/10.1109/TGRS.2025.3539532, 2025.
- Kelly, R., Chang, A. T., Tsang, L., and Foster, J. L.: A prototype AMSR-E global snow area and snow depth algorithm, IEEE Transactions on Geoscience and Remote Sensing, 41, 230–242, https://doi.org/10.1109/TGRS.2003.809118, 2003.
- Lievens, H., Demuzere, M., Marshall, H. P., Reichle, R. H., Brucker, L., Brangers, I., de Rosnay, P., Dumont, M., Girotto, M., Immerzeel, W. W., Jonas, T., Kim, E. J., Koch, I., Marty, C., Saloranta, T., Schöber, J., and De Lannoy, G. J. M.: Snow depth variability in the Northern Hemisphere mountains observed from space, Nat Commun, 10, https://doi.org/10.1038/s41467-019-12566-y, 2019.
- Meloche, J., Royer, A., Langlois, A., Rutter, N., and Sasseville, V.: Improvement of microwave emissivity parameterization of frozen Arctic soils using roughness measurements derived from photogrammetry, Int J Digit Earth, 14, 1380–1396, https://doi.org/10.1080/17538947.2020.1836049, 2021.

- Pulliainen, J. and Grandeil, J.: HUT snow emission model and its applicability to snow water equivalent retrieval, IEEE Transactions on Geoscience and Remote Sensing, 37, 1378–1390, https://doi.org/10.1109/36.763302, 1999.
- Saberi, N., Kelly, R., Toose, P., Roy, A., and Derksen, C.: Modeling the observed microwave emission from shallow multi-layer Tundra Snow using DMRT-ML, Remote Sens (Basel), 9, https://doi.org/10.3390/rs9121327, 2017.

---

## Referee Report (RR1)

**Revisions of: A Prototype Passive Microwave Retrieval Algorithm**

for Tundra Snow Density

By: Welch and Kelly

Benoit Montpetit

**1 General Comments**

I will start by saying how impressed I am about the amount of quality work that was done for this review. Most comments from the reviewers have been addressed and this study is now much more rigorous.

The radiative transfer modelling has considerably been improved. This includes switching to IBA instead of non-sticky spheres DMRT. This allowed to work with SSA as a grain size proxy and the evolution of SSA is well documented in the Arctic. The authors use good models to treat the grain growth. Including atmospheric corrections like what is proposed in the GlobSnow products also improves the rigorousness of the study. Allowing the depth hoar layer thickness to evolve in the calibration process also brings great value to this study. Showing that having this parameter as a dynamic variable, allows to achieve the desired uncertainty range ( $\pm 10\%$ ), is a great result in itself.

Although I am not fond of retrievals done at the PMW scale, I do appreciate the discussion section, which takes it into account, and shows the usefulness of the methodology with its limitations.

I only have a few minor edits/suggestions listed below. NOTE: the line number are associated with the track-change document.

- L. 15: ... parameterization was used ...
- 15 L. 49: delete "in"
  - L. 51: ... the difficulty in ... L. 64: delete "in" before intensive
  - L. 163: explicitly mention "wind" slab layer. Something that needs to be done throughout the manuscript
  - L. 247: AMSR-E
  - L. 302: microwave-plausible
- 20 L. 315: I feel like section 4.4 should be in the data section since it describes stages of snowpack evolution
  - L. 376: Domine et al., (2018), the whole reference should not be in ()

- L. 382: Add o symbols throughout section 4.4.3
- L. 392: metamorphoses
- L. 394: The Taillandier et al. relationship has been shown to work well for Arctic snowpacks, good choice!
- 25 L. 411: microwave-plausible, please review the entire manuscript to ensure it is all switched from DMRT-plausible
  - L. 466: polar desert
  - L. 489: compared to those
  - L. 492: great results with regards to the depth hoar layer impact!
  - L. 516: simulations are included
- 30 Figure 7: Very important to add the SSA curves here. It will allow the reader to truly understand what is happening in the retrieval process. Please do the same in the figures of the appendix.
  - L. 600: Detailed snow surveys
  - L. 607: oasis
  - L. 668: causing

35

Finally, when submitting track-change documents, I encourage the authors to take the time and remove comments and submit a clean document for review.